# Perioperative Complications after Hip and Knee Revision Arthroplasty in the over 80 Years Old Population: A Retrospective Observational Case–Control Study

**DOI:** 10.3390/jcm12062186

**Published:** 2023-03-11

**Authors:** Vincenzo Di Matteo, Marina Di Pilla, Francesco La Camera, Emanuela Morenghi, Guido Grappiolo, Mattia Loppini

**Affiliations:** 1IRCCS Humanitas Research Hospital, Via Alessandro Manzoni 56, 20089 Rozzano, Milan, Italy; 2Adult Reconstruction and Joint Replacement Service, Division of Orthopaedics and Traumatology, Fondazione Policlinico Universitario Agostino Gemelli IRCCS, Largo Agostino Gemelli 8, 00168 Roma, Italy; 3Faculty of Medicine and Surgery, Catholic University of Sacred Heart, Largo Francesco Vito 1, 00168 Rome, Italy; 4Fondazione Livio Sciutto Onlus, Campus Savona, Università degli Studi di Genova, Via Magliotto 2, 17100 Savona, Italy; 5Department of Biomedical Sciences, Humanitas University, Via Rita Levi Montalcini 4, 20090 Pieve Emanuele, Milan, Italy

**Keywords:** hip, knee, octogenarians, perioperative complication, readmission rate, revision arthroplasty

## Abstract

Background: The number of joint revision arthroplasties has increased in the elderly population, which is burdened by several perioperative risks. Methods: Patients who underwent hip and knee revision arthroplasty were retrospectively included, and they were divided into two groups by age: <80 years old (Group 1) and ≥80 years old (Group 2). The primary outcome was to compare perioperative complication rates. The secondary outcome was to compare the 30-day, 90-day, and 1-year readmission rates. Results: In total, 74 patients in Group 1 and 75 patients in Group 2 were included. Postoperative anemia affected 13 patients in Group 1 (17.6%) and 25 in Group 2 (33.3%, *p* 0.027); blood units were transfused in 20 (26.7%) and 11 (14.9%, *p* 0.076) patients, respectively. In Group 1, two (2.7%) patients reported wound infection. In Group 2, eight (10.7%) patients presented hematomas, and two (2.7%) patients reported dislocations. No significant differences in the two groups were observed for 30-day (*p* 0.208), 90-day (*p* 0.273), or 1-year readmission rates (*p* 0.784). Conclusion: The revision arthroplasty procedure in patients over 80 years old is not associated with a higher risk of perioperative complications, or higher readmission rate compared with younger patients undergoing hip and knee revision surgery.

## 1. Introduction

Hip and knee arthroplasties are the most commonly performed elective orthopedic procedures in the over 65 years old population [1]. The number of geriatric patients, defined as people more than 80 years old, is going to increase. As the life expectancy of patients is increasing, more elderly patients are undergoing THA (total hip arthroplasty) and TKA (total knee arthroplasty) due to increasing incidence of osteoarthritis in this age group [2]. Recent advances in health care have enabled these patients to undergo major procedures such as TKA and THA, helping them to pursue more active lifestyles. The longer life expectancy and the higher quality of life of active patients who have undergone total joint replacement leads to the wearing out of various implants and implant failure for various reasons [3]. One of the main components that becomes overloaded and prone to wear is polyetylene; the analysis of tribology and the selection of couplings has a strategic role to improve performance and implant survival [4,5]. Consequently, as patients age and their prosthetic implants wear out, the number of revision arthroplasties in the elderly population over 80 years old performed by orthopedic surgeons will increase [1,6,7]. Advancing age is reported to be a significant risk factor in the incidence of in-hospital complications [6]. Although hip and knee revision arthroplasties in geriatric patients are burdened by a considerable risk, especially in those affected by multiple comorbidities [8,9,10,11,12], they may offer benefits in improving functional independence and the quality of life, consequently increasing patients’ life expectancy [13]. Awareness of a higher incidence of major systemic complications should alert the treating surgeon to carry out comprehensive perioperative management in this subset of patients, which could lead to better outcomes [14].

Although a few studies have already investigated perioperative complications after total hip and knee revision arthroplasty in the elderly [15,16], to the best of our knowledge, only one compared the rate of perioperative complications following aseptic revision of THA in patients aged ≥ 80 with that of those <80 [17]. Because it was based on a national database in the United States, some bias affected the study, such as different perioperative management, application of drug protocols, types of implants, and number of procedures per surgeon. Therefore, further independent cohort studies are required to confirm these findings. The primary outcome of this study was to determine perioperative local and systemic complications after hip and knee revision surgery in a population ≥ 80 years compared to those <80 years. The secondary outcome was to determine the 30-day, 90-day, and 1-year readmission rates after discharge from the hospital in the two groups.

## 2. Materials and Methods

### 2.1. Protocol

The present retrospective observational case–control study used medical file records of patients included in a registry of orthopedic surgical procedures. The study included 149 patients undergoing hip or knee revision arthroplasty by senior surgeons experienced in joint replacement surgery, from 1 February 2017 to 31 July 2020 at the Humanitas Research Hospital IRCCS, Italy. Patients were identified from hospital clinical records using International Classification of Diseases, Ninth Revision, Clinical Modification (ICD9-CM) procedure codes (00.70, 00.71,00.72, 00.73, 00.74, 00.75, 00.76, 00.77 for revision THA and 80.06,81.55, 00.80, 00.81, 00.82, 00.83, 00.84 for revision TKA). The patients included were divided into two groups based on age at the time of surgery: < 80 years old (Group 1) and those geriatrics ≥80 years old (Group 2). Participant exclusion criteria were sequelae of periprosthetic infections, malignancy, patients under 18 years old, and less than 1 year of follow-up. Patients were matched by gender, diseased joint, preoperative diagnosis, type, and period of surgery.

Femoral stem revision was performed using uncemented monoblock splined tapered gritblasted stems (Wagner-SL stem; Zimmer, Warsaw, Indiana, US) or uncemented Arcos Modular Femoral Revision System (Zimmer Biomet Inc., Warsaw, Indiana, US) in cases of major defect in order to avoid leg length discrepancy (LLD). Acetabular revision was performed using Trabecular Metal shell (TM Modular, Trilogy TM and augment or TM Revision shell, Zimmer Biomet, Warsaw, Indiana, US); for major acetabular defect, a dual mobility cup (DMC Avantage; Zimmer Biomet, Warsaw, Indiana, US) cemented into a larger shell was used. Revision knee arthroplasty was performed using NexGen^®^ Legacy^®^ Constrained Condylar Knee (LCCK) with or without Trabecular Metal™ Cones augments (Zimmer Biomet, Warsaw, Indiana, US).

Data regarding patient demographic characteristics, body mass index (BMI) in kg/m^2^, clinical variables, American Society of Anesthesia (ASA) classification, comorbid conditions, perioperative medical complications, length of stay (LOS), use of thromboprophylaxis, and inflammatory indicators were obtained from medical file records. During hospitalization, all complications and their management were documented in the clinical file records that were analyzed for the purposes of this study. Perioperative complications that emerged from the first perioperative day to the day before discharge home or transfer to the rehabilitation department were recorded. Perioperative local complications are those occurring at the level of the surgical site, including hematomas, superficial infections, deep infections, fractures, dislocations, and contact dermatitis associated with compression stockings. Perioperative systemic complications include deep vein thrombosis, pulmonary embolism and other pulmonary complications, gastrointestinal disorders, and neurological and genitourinary complications. The occurrence of postoperative anemia was measured: the postoperative Hb (hemoglobin) values, the number of patients who underwent blood transfusions, and the number of blood units transfused were recorded. Postoperative anemia is defined by a postoperative Hb ≤ 8 g/dL on the third day after surgery or blood transfusions performed within the second postoperative day. The threshold values for transfusing were based on the PBM (Patient Blood Management) guidelines: Hb < 7 g/dL or Hb < 8 g/dL in cardiopathic or symptomatic patients (thoracic pain, orthostatic hypotension, tachycardia) [18]. The 30-day, 90-day, and 1-year readmission rates of these two groups were recorded.

### 2.2. Statistical Analysis

Data were described as number and percentage for categorical variables, or mean and standard deviation for continuous variables. Differences between Group 1 and Group 2 were explored with the chi-square test if categorical, with Student’s *t* test if continuous Gaussian distributed, or with Mann–Whitney if otherwise. Association between risk factors and early mobilization at day 0 and risk factors and postoperative anemia were explored with univariable logistic regression. All independent factors with a *p* value under 0.1 were then submitted to a backward multivariable logistic regression. An explorative analysis was performed for association of age groups and for systemic complications, 30-day, 90-day, and 1-year readmission rate. A *p* value under 0.05 was considered significant. All analyses were performed with stata15.

## 3. Results

### 3.1. Selection of Study Population

Patients were divided into two groups based on age at the time of surgery. Demographic results are reported in Table 1. A total of 312 patients were identified; 266 patients underwent revision THA, and 46 underwent revision TKA. Among 266 patients who underwent RTHA (revision total hip arthroplasty), 118 were affected by periprosthetic infections, and 20 were lost during follow-up. Therefore, only 128 patients were eligible for the study: 66 patients ≥ 80 years old, and 62 < 80 years old. Among 46 patients who underwent RTKA (revision total knee arthroplasty), 19 were periprosthetic infections, and 6 were lost during follow-up. Therefore, only 21 patients were eligible for the study: 9 patients ≥ 80 years old and 12 < 80 years old. The flow chart of the patient selection and inclusion process is shown in Figure 1.

Group 1 (<80 years old) included 74 patients with a mean age of 60.5 (18–78, σ 2.3) years old: there were 34 male (45.95%) and 40 female patients (54.1%); 62 (83.8%) underwent hip revision, and 12 (16.2%) underwent knee revision arthroplasty.

Group 2 (≥80 years old) included 75 patients with a mean age of 82.5 (80–89, σ 11.9) years old: there were 34 male (45.3%) and 41 female patients (54.7%); 66 (88.0%) underwent hip revision, and 9 (12.0%) underwent knee revision arthroplasty.

### 3.2. Characteristics of Study Population

The two groups significantly differed for some comorbidities (Table 1): arterial hypertension (44.6% and 68.0%, respectively, in Group 1 and 2, *p* 0.004); cardiovascular diseases, specifically coronary artery disease and atrial fibrillation (10.8% and 29.3%, respectively, in Group 1 and 2, *p* 0.005); carotid artery stenosis (8.1% and 24%, respectively, in Group 1 and 2, *p* 0.013); and preoperative anemia (10.8% and 29.3%, respectively, in Group 1 and 2, *p* 0.005). The ASA (American Society of Anesthesiologists) score for Group 1 and Group 2 was I in 74.3% and 38.7%; II in 24.3% and 49.3%; and III in 1.4% and 12.0%, respectively (*p* < 0.001). Moreover, serum levels of iron differed significantly in the two groups (86.6 ± 32.7 vs. 73.6 ± 30.0, respectively, in Group 1 and 2, *p* 0.016). Group 1 mean length of stay (LOS) was 5.4 days (3–16, σ 2.5), and in Group 2, respectively, it was 5.6 days (4–16, σ 2.7), *p* 0.833.

Perioperative local complications (Figure 2) observed during hospitalization were postoperative anemia, need of blood transfusions, hematomas, superficial infections of surgical site, and dislocations. Postoperative anemia affected 13 patients in Group 1 (17.6%) and 25 in Group 2 (33.3%) (*p* 0.027). Blood units were transfused in 11 patients (14.9%) of Group 2 and 20 patients (26.7%) of Group 1, but with no significant difference (*p* 0.076). No hematomas were found in Group 1, but 8 patients (10.7%) presented hematomas in Group 2. Wound infection was observed in 2 patients only in Group 1 (2.7%), and 2 dislocations happened only in Group 2 (2.7%).

Perioperative systemic complications in the two groups were recorded in Group 1 and Group 2 (Figure 3): 6 patients (8.1%) and 4 patients (5.3%), respectively, were diagnosed with fever, and 2 patients (2.7%) and 5 patients (6.7%), respectively, were diagnosed with urinary tract infections. In Group 2 only 1 patient was diagnosed with a thromboembolic event (1.3%), 7 patients needed low-flow oxygen therapy (9.3%), 5 patients were diagnosed with new-onset atrial fibrillation and acute myocardial infarction (6.7%), 7 patients were diagnosed with postoperative delirium (9.3%), and only 1 patient was diagnosed with a neurological complication (1.3%).

No significant differences in the two groups were observed for the hospital readmission rate at 30 days, 90 days, and 1 year (Figure 4). Readmission at 30 days occurred in 1 patient (1.4%) in Group 1 and 5 patients (6.7%) in Group 2 (*p* 0.208); 90-day readmission occurred in 2 patients (2.7%) in Group 1 and 6 patients (8.0%) in Group 2 (*p* 0.273); and 1-year readmission occurred in 8 patients (10.8%) in Group 1 and 7 patients (9.3%) in Group 2 (*p* 0.784).

Factors significantly associated with early mobilization after surgery are summarized in Table 2. After multivariable analysis, we observed that early mobilization at day 0 was significantly associated only with preoperative anemia (OR = 5.67, 95% CI 1.63–19.77, *p* 0.006). The association of risk factors with postoperative anemia is shown in Table 3. After multivariable analysis, postoperative anemia results were associated with preoperative anemia (OR = 8.34, 95% CI 3.24–21.50, *p* < 0.001) and osteoporosis (OR = 4.10, 95% CI 1.35–12.45, *p* 0.013). No evidence of association was observed between blood transfusion and antiplatelet therapy (18.0% vs. 21.8% of blood transfusion in antiplatelet therapy or not, respectively, *p* 0.609) or anticoagulant therapy (23.1% vs. 20.6% of blood transfusion in anticoagulant therapy or not, respectively, *p* 0.734).

## 4. Discussion

The main finding of the present study was that patients ≥ 80 years old who underwent hip and knee revision arthroplasty reported a low rate of perioperative local complications compared to the younger control group. On the other hand, perioperative systemic complications were higher in the elderly population compared to younger patients. Moreover, the older age of patients was not associated with a longer length of stay in the hospital, and it was not related to a higher risk of readmission at 30 days, 90 days, and 1 year after surgery.

The higher risk of urinary tract infections identified can be a target to improve medical care quality. These infections are associated with indwelling catheters, and early removal of the catheters must be recommended. The increased risk of respiratory complications such as postoperative pneumonia in patients ≥ 80 years old compared to younger patients must be highlighted. Postoperative preventive measures such as chest physiotherapy for pneumonia and breathing exercises must be recommended to reduce pneumonia rates, especially in older patients. The high risk of cardiovascular complications in older patients should be treated with medication preoperatively and continued throughout the postoperative period. This study identified risk factors significantly correlated to early mobilization and verticalization: preoperative anemia was the only factor significantly associated with time of verticalization the day of surgery. Patients who suffered from preoperative anemia were treated with iron supplementation (infusion or oral therapy) to increase their iron deposits before surgeries procedures, resulting in an even better postoperative iron profile than nonanemic patients. Unfortunately, this was not supported by laboratory data since not all patients who underwent iron supplementation had repeated preoperative blood tests. Furthermore, multivariable analysis assessing significant associations between risk factors and postoperative anemia identified preoperative anemia as a significant risk factor for postoperative anemia. RTHA and RTKA were safe for patients ≥ 80 years old in our high-volume single center, where all surgeries were performed by senior surgeons experienced in joint replacement surgery. Furthermore, the co-management between the surgeon and the hospitalist, a doctor in the field of internal medicine, during the patient’s hospitalization potentially reduced complications, improving medical care [14,19,20,21,22,23].

The majority of previous studies in the geriatric population, especially case–control studies, were conducted on primary hip and knee arthroplasty and very few on total hip and knee revision arthroplasty. Bovonratwet et al., contrary to our results, reported greater risks in patients aged ≥ 80 years compared with younger patients after aseptic RTHA and highlighted the need for medical optimization in these vulnerable patients [17]. Dugdale et al. found, contrary to our results, that nonagenarian patients had longer length of stay (LOS), higher total charges, and were more likely to develop several more major complications than octogenarians following both PTHA (primary total hip arthroplasty) or RTHA [15]. Birdsall et al. reported a mortality rate of 5% after a knee revision arthroplasty of 119 patients aged ≥ 80 years old [24]. Smith et al. compared perioperative complications after PTKA (primary total knee arthroplasty) and RTKA between octogenarians and nonagenarians. The investigators observed a higher risk of perioperative complications, such as postoperative anemia and need of blood transfusions, in the older patient group [25]. Jauregui et al. have demonstrated the cost–benefit ratio of both primary and revision arthroplasty for the geriatric population: on one hand, octogenarians had a higher risk of perioperative complications which prolong hospital stay and sanitary costs; on the other hand, a good functional outcome and pain control emerged regardless of age [26]. The investigators observed a significantly higher rate of hospital readmission for all causes in the elderly population [15,27,28,29,30,31,32], especially dislocations, which occur in up to 20% of older patients [33,34,35]. Parvizi et al. have demonstrated that preexisting cardiovascular diseases represent a risk factor for intraoperative mortality in over 80 years old patients and a 30-day mortality rate of 0.29% in all age groups [36,37]. Birdsall et al. reported a 1-year mortality rate of 5% in octogenarians [24]. Respiratory and urinary tract infections and delirium are more statistically significant in the older population [38]. Smith et al. compared one group of nonagenarians with one group of octogenarians who underwent either primary or revision arthroplasty: the older group reported a higher risk of needed blood transfusion, urinary tract infection, and acute kidney injury than the younger group [25]. Increases in life expectancy and implant failures have resulted in more elderly patients requiring revision THA and TKA [39,40]. Revision surgery has been projected to increase by 137% between 2005 and 2030 [2]. Previous studies have shown revision arthroplasty to be effective in reducing pain and in increasing function and quality of life in elderly patients.

Wear is the main reason for hip and knee revision arthroplasty [41,42]. Reducing wear is a strategic step in minimizing harmful failures for users. Metal-on-metal bearings also carry the danger of poisoning due to metallosis. The main drawback of this type of bearing is that worn metal particles can spread through the lymphatic system to locations far from the implant, and it has been reported that the metals can accumulate in the liver, spleen, lymph nodes, and bone marrow. Along with the reactive nature of worn metal particles, they have the potential to create local inflammation, contributing to the emergence of osteolysis, or cause cytotoxicity, hypersensitivity, and neoplasia [43,44]. Jamari et al. reported that dimples can reduce contact pressure and wear, indicating that in real use they could reduce failure due to wear and poisoning due to metallosis. The bottom profile dimple ball type is estimated to produce the lowest wear compared to other models; this model demonstrated a promising surface textured parameter and could be used to design the bearing components in a total hip arthroplasty [45].

This study has several limitations. First, it is a retrospective observational study; as with any database, the quality of data and missing data may introduce errors. Second, the study was performed in a high-volume single center with a high level of expertise in elective prosthetic surgery whose results cannot be generalized. Third, the preoperative schedule selection bias selected very healthy patients or patients with stable chronic diseases; therefore, the conclusion of the study may be applicable only to patients with low comorbidities. The main strength of this study is the large number of patients enrolled. Second, it is the only retrospective study that has evaluated perioperative complications and readmission rates after hip and knee revision surgery in patients ≥ 80 years old compared to patients < 80 years old in a high-volume single center. Further research would be necessary to determine the clinical relevance of the higher perioperative systemic complications found in nonagenarian and octogenarian patients, highlighting the need of medical optimization and post-discharge planning following total hip and knee revision arthroplasty.

## 5. Conclusions

In conclusion, 80-year-old patients are not associated with a higher risk of perioperative complications or readmission rate compared with younger patients undergoing hip and knee revision surgery. Therefore, age could not be considered an absolute contraindication to revision arthroplasty procedures.

## Figures and Tables

**Figure 1 jcm-12-02186-f001:**
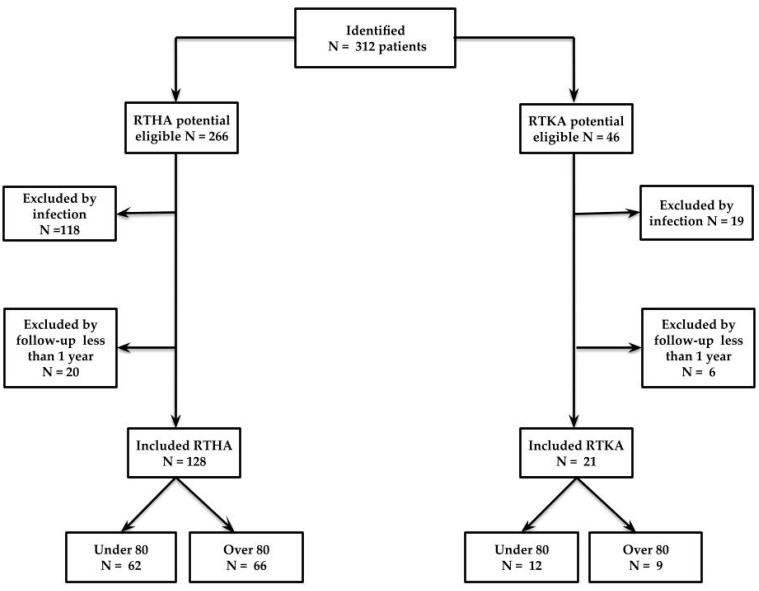
Flow chart of the patient selection and inclusion process. RTHA, revision total hip arthroplasty; RTKA, revision total knee arthroplasty.

**Figure 2 jcm-12-02186-f002:**
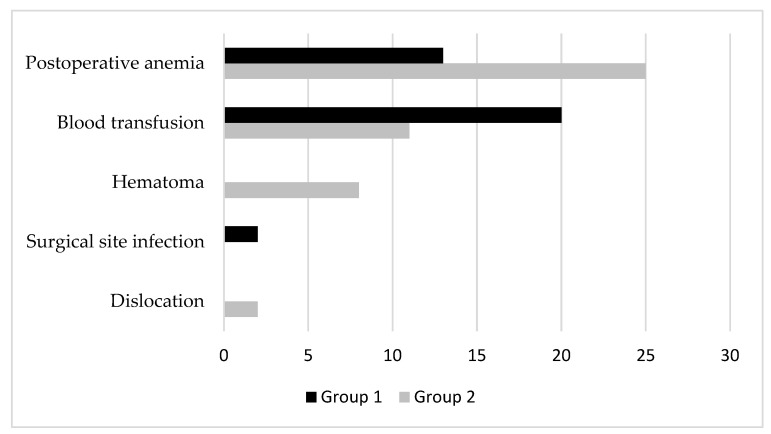
Perioperative local complications in the two groups.

**Figure 3 jcm-12-02186-f003:**
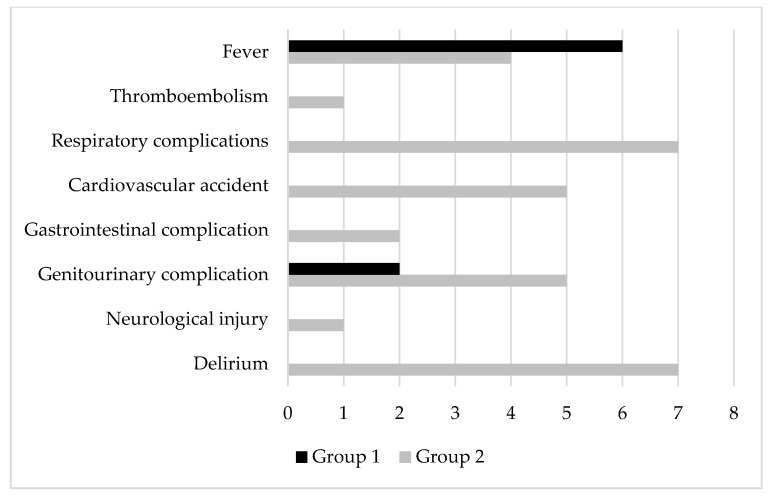
Perioperative systemic complications in the two groups.

**Figure 4 jcm-12-02186-f004:**
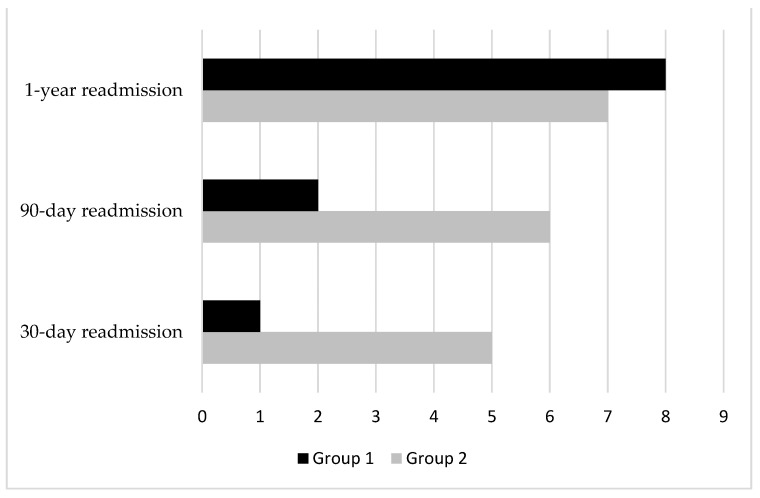
30-day, 90-day, and 1-year hospital readmission rates in the two groups.

**Table 1 jcm-12-02186-t001:** Characteristics of the patient population in the two groups.

	Group 1 Age < 80	Group 2 Age ≥ 80	*p*
N	74	75	
Male (%)	34 (45.95%)	34 (45.3%)	0.826
BMI	25.9 ± 4.3	26.0 ± 4.4	0.612
LOS (n days)	5 (3–16)	5 (3–16)	0.833
Smoking	9 (12.16%)	5 (6.67%)	0.276
Arterial hypertension	33 (44.59%)	51 (68.00%)	0.004
Dyslipidemia	24 (32.43%)	22 (29.33%)	0.682
Impaired fasting glycemia	4 (5.41%)	8 (10.67%)	0.367
Diabetes mellitus II	9 (12.16%)	4 (5.33%)	0.156
Cardiovascular diseases	8 (10.81%)	22 (29.33%)	0.005
Disthyroidism	13 (17.57%)	10 (13.33%)	0.474
COPD	4 (5.41%)	9 (12.00%)	0.245
Gastrointestinal diseases	3 (4.05%)	9 (12.00%)	0.130
Liver diseases	2 (2.70%)	8 (10.67%)	0.098
Chronic kidney diseases	4 (5.41%)	12 (16.00%)	0.061
Carotid artery stenosis	6 (8.11%)	18 (24.00%)	0.013
Preoperative anemia	8 (10.81%)	22 (29.33%)	0.005
Osteoporosis	9 (12.16%)	14 (18.67%)	0.272
Hematologic diseases	1 (1.35%)	5 (6.67%)	0.209
Rheumatologic diseases	5 (6.76%)	7 (9.33%)	0.765
Active neoplasia	0	4 (5.33%)	0.120
Depressive anxiety syndrome	9 (12.16%)	14 (18.67%)	0.272
Venous thromboembolism	7 (9.46%)	5 (6.67%)	0.531
Hemorrhagic diseases	1 (1.35%)	3 (4.00%)	0.620
Ischemic stroke/TIA	5 (6.76%)	5 (6.67%)	1.000
Cognitive impairment	0	3 (4.00%)	0.245
Parkinsonism	1 (1.35%)	6 (8.00%)	0.116
ASA			<0.001
1	55 (74.32%)	29 (38.67%)	
2	18 (24.32%)	37 (49.33%)	
3	1 (1.35%)	9 (12.00%)	
Ongoing antiplatelet therapy	16 (21.62%)	23 (30.67%)	0.209
Ongoing anticoagulant therapy	3 (4.05%)	10 (13.33%)	0.078
Preoperative Hb (g/dL)	13.9 ± 1.5	12.9 ± 1.5	<0.001
Preoperative creatinine (mg/dL)	0.75 ± 0.23	0.87 ± 0.35	0.152
Preoperative C-reactive protein (mg/dL)	0.75 ± 1.46	1.32 ± 3.36	0.056
Ferritin (ng/mL)	93.1 ± 78.7	115.3 ± 93.4	0.169
Serum iron (mcg/dL)	86.6 ± 32.7	73.6 ± 30.0	0.016

BMI, body mass index; LOS, length of stay; COPD, chronic obstructive pulmonary disease; TIA, transient ischemic attack; ASA, American Society of Anesthesiologists; Hb, hemoglobin.

**Table 2 jcm-12-02186-t002:** Univariable and multivariable logistic regression assessing risk factors for early mobilization at day 0.

	Univariable		Multivariable	
	OR (95% CI)	*p*	OR (95% CI)	*p*
Age (<80)	0.50 (0.25–1.00)	0.050	0.65 (0.31–1.35)	0.247
Gender (F)	1.34 (0.66–2.71)	0.417		
BMI	1.03 (0.65–1.11)	0.541		
LOS	1.01 (0.88–1.15)	0.891		
Smoking	0.87 (0.28–2.75)	0.813		
Arterial hypertension	1.38 (0.69–2.75)	0.357		
Dyslipidemia	1.36 (0.64–2.91)	0.423		
Impaired fasting glucose/Diabetes mellitus	2.02 (0.70–5.78)	0.191		
Cardiovascular diseases	1.45 (0.59–3.53)	0.419		
Dysthyroidism	1.14 (0.44–2.99)	0.786		
COPD	0.55 (0.17–1.72)	0.301		
Gastrointestinal disease	2.61 (0.55–12.41)	0.227		
Liver disease	0.72 (0.19–2.67)	0.621		
Chronic kidney disease	2.29 (0.62–8.45)	0.213		
Carotid artery stenosis	2.06 (0.72–5.91)	0.177		
Preoperative anemia	5.67 (1.63–19.77)	0.006	5.67 (1.63–19.77)	0.006
Osteoporosis	0.72 (0.29–1.81)	0.489		
Hematologic disease	2.53 (0.29–22.23)	0.404		
Rheumatologic diseases	0.46 (0.14–1.50)	0.197		
Active neoplasia	NC			
Depressive anxiety syndrome	1.93 (0.67–5.56)	0.222		
Venous thromboembolism	2.61 (0.55–12.41)	0.227		
Hemorrhagic disease	NC			
Ischemic stroke/TIA	0.46 (0.13–1.68)	0.242		
Cognitive impairment	NC			
Parkinsonism	1.24 (0.23–6.61)	0.804		
ASA score	1.47 (0.82–2.63)	0.192		
Antiplatelet therapy	1.14 (0.52–2.50)	0.743		
Anticoagulant therapy	6.55 (0.83–51.88)	0.075	5.08 (0.62–41.84)	0.130
Preoperative anemia	0.69 (0.53–0.88)	0.003		
Preoperative creatinine	2.23 (0.61–8.16)	0.223		
Preoperative C-reactive protein	1.12 (0.88–1.41)	0.363		
Ferritin	1.00 (1.00–1.01)	0.349		
Serum iron level	1.00 (0.99–1.01)	0.619		

BMI, body mass index; LOS, length of stay; COPD, chronic obstructive pulmonary disease; TIA, transient ischemic attack; ASA, American Society of Anesthesiologists.

**Table 3 jcm-12-02186-t003:** Univariable and multivariable logistic regression assessing risk factors for postoperative anemia.

	Univariable		Multivariable	
	OR (95% CI)	*p*	OR (95% CI)	*p*
Age (<80)	0.43 (0.20–0.91)	0.029	0.62 (0.24–1.63)	0.333
Gender (F)	1.13 (0.52–2.44)	0.763		
BMI	0.93 (0.85–1.02)	0.142		
LOS	1.13 (0.99–1.29)	0.064		
Smoking	0.78 (0.21–2.96)	0.714		
Arterial hypertension	0.61 (0.29–1.29)	0.196		
Dyslipidemia	0.74 (0.33–1.70)	0.482		
Impaired fasting glucose/Diabetes mellitus	2.45 (0.98–6.11)	0.055		
Cardiovascular disease	1.08 (0.43–2.68)	0.870		
Dysthyroidism	0.57 (0.18–1.80)	0.337		
COPD	0.22 (0.03–1.76)	0.154		
Gastrointestinal disease	2.25 (0.67–7.57)	0.190		
Liver disease	0.72 (0.15–3.53)	0.681		
Chronic kidney disease	1.38 (0.45–4.26)	0.578		
Carotid artery stenosis	1.58 (0.62–4.06)	0.339		
Preoperative anemia	7.43 (3.10–17.80)	<0.001	8.34 (3.24–21.50)	<0.001
Osteoporosis	2.69 (1.07–6.79)	0.036	4.10 (1.35–12.45)	0.013
Hematologic disease	3.09 (0.60–15.99)	0.179		
Rheumatologic disease	1.51 (0.43–5.35)	0.519		
Active neoplasia	3.03 (0.41–22.28)	0.277		
Depressive anxiety syndrome	1.71 (0.66–4.42)	0.271		
Venous thromboembolism	0.25 (0.03–1.97)	0.186		
Hemorrhagic disease	NC			
Ischemic stroke/TIA	2.06 (0.55–7.73)	0.285		
Cognitive impairment	NC			
Parkinsonism	1.18 (0.22–6.34)	0.849		
ASA score	1.68 (0.94–2.99)	0.079	1.06 (0.49–2.33)	0.877
Antiplatelet therapy	0.84 (0.36–1.97)	0.686		
Anticoagulant therapy	1.33 (0.39–4.61)	0.649		
Preoperative anemia	0.51 (0.39–0.68)	<0.001		
Preoperative creatinine	1.62 (0.50–5.27)	0.425		
Preoperative C-reactive protein	1.22 (0.98–1.51)	0.075	1.05 (0.86–1.29)	0.619
Ferritin	1.004 (1.000–1.009)	0.033		
Serum iron level	0.99 (0.98–1.00)	0.162		

BMI, body mass index; LOS, length of stay; COPD, chronic obstructive pulmonary disease; TIA, transient ischemic attack; ASA, American Society of Anesthesiologists.

## Data Availability

The data supporting reported results can be found in a repository (Zenodo).

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
