# Peer review of "Perioperative Complications after Hip and Knee Revision Arthroplasty in the over 80 Years Old Population: A Retrospective Observational Case–Control Study"

_jcm, 2023, doi:10.3390/jcm12062186_

Round 1

Reviewer 1 Report

1.      The abstract requires the addition of quantitative results.

2.      Given the “take-home” message at the end of the abstract, the present form was insufficient.

3.      Put the keywords in a new order based on alphabetical order.

4.      What is the novel of the present study? It works have been widely studied in the past. Nothing something really new in the present form. The lack of novel seems to make the present study like to replication/modified study. The authors need to detail their novelty in the introduction section. It is a major concern for rejecting this paper.

5.      Previous study related needs to explain in the introduction section consisting of their work, their novelty, and their limitations to show the research gaps that intend to be filled in the present study.

6.      In line 37-54, the authors explain about hip arthroplasty is need more additional relevant literature to support the explanation. The MDPI's suggested reverence should be applied in the explanation as follows: Minimizing Risk of Failure from Ceramic-on-Ceramic Total Hip Prosthesis by Selecting Ceramic Materials Based on Tresca Stress. Sustainability 2022, 14, 13413. https://doi.org/10.3390/su142013413

7.      In the materials and methods, the authors need to add additional illustrations as a form of figure that explains the workflow of the present study to make the reader easier to understand rather than only the dominant text as a present form.

8.      What is the baseline of patient selection? Is there any protocol, standard, or basis that has been followed? It is unclear since the patient is very heterogeneous with a small number. The resonance involved impacts the present result makes this study flaws. One major reason for rejecting this paper.

9.      It's also essential to include additional information on the manufacturer, country, and specifications of the tools.

10.   The error and tolerance of the experimental tools used in this investigation are important aspects that have to be mentioned in the manuscript. It might be valuable for further research by other scholars because of the different results.

11.   Results must be compared to similar past research.

12.   The authors need to improve the discussion in the present article become more comprehensive. The present form was insufficient.

13.   Please include the limitation of the present study, it is missing.

14.   Information to the conclusion by formatting it as a paragraph rather than the manner as point-by-point.

15.   The conclusion section needs to explain further research.

16.   The authors need to enrich the reference from five years back. MDPI reference is strongly recommended.

17.   The authors were encouraged to proofread their work due to grammatical problems and linguistic style.

18.   It is suggested to the authors for providing graphical abstract in the system after revision.

Author Response

Manuscript ID: jcm- 2160996

Type of manuscript:

Article Title: Perioperative complications after hip and knee revision arthroplasty in the over 80 years old population: a retrospective observational case-control study.

Dear Reviewer 1,

Please find attached the revised version of the above manuscript. Your comments of the reviewers have been carefully considered, and implemented as follows. Please note that all changes made in the document have been highlighted in yellow to facilitate tracking and reading.

Reviewer 1

  1. The abstract requires the addition of quantitative results.

We added quantitative results in the abstract including the total number of patients for each group and the readmission rates (lines 19-24).

  1. Given the “take-home” message at the end of the abstract, the present form was insufficient.

We modified the conclusion at the end of the abstract including the take-home message (lines 24-26).

  1. Put the keywords in a new order based on alphabetical order.

We changed the order as required (lines 29-29).

  1. What is the novel of the present study? It works have been widely studied in the past. Nothing something really new in the present form. The lack of novel seems to make the present study like to replication/modified study. The authors need to detail their novelty in the introduction section. It is a major concern for rejecting this paper.

Although a few studies have already investigated the perioperative complication after revision total hip arthroplasty and totale knee arthroplasty in the elderly, to the best of our knowledge, only one compare the rate of perioperative complications following aseptic revision in THA (RTHA only) in patients aged > 80 with that those < 80 [1]. Because it was based on National database in the United States, some bias affected it such as the different perioperative management, application of the drug protocols, types of implants, and number of procedures per surgeon. Therefore, further independent cohort studies are required to confirm these findings. The aim of the present study was to determine evaluates postoperative local and systemic complications after hip and knee revision surgery in a population older than 80 years compared to those younger 80 years; analyzes patient’s risk factors; and reports the 30-day, 90-day, and 1-year readmission rates after discharge from hospital in a high-volume single center.

[1] Bovonratwet, P.; Malpani, R.; Ottesen, T.D.; Tyagi, V.; Ondeck, N.T.; Rubin, L.E.; Grauer, J.N. Aseptic Revision Total Hip Arthroplasty in the Elderly: Quantifying the Risks for Patients over 80 Years Old. The Bone & Joint Journal 2018, 100-B, 143–151, doi:10.1302/0301-620X.100B2.BJJ-2017-0895.R1.

  1. Previous study related needs to explain in the introduction section consisting of their work, their novelty, and their limitations to show the research gaps that intend to be filled in the present study.

Different past studies about revision procedures are controversial, with small sample sizes from over a decade ago, also lacked data about the 30-day readmission rate and elderly patient’s specific risk factors for complications. This information has become especially important in the present climate of increasing expectations, bundled payments, pay for performance and comparison of hospital metrics.

  1. In line 37-54, the authors explain about hip arthroplasty is need more additional relevant literature to support the explanation. The MDPI's suggested reverence should be applied in the explanation as follows: Minimizing Risk of Failure from Ceramic-on-Ceramic Total Hip Prosthesis by Selecting Ceramic Materials Based on Tresca Stress. Sustainability 2022, 14, 13413. https://doi.org/10.3390/su142013413.

We modified the statement highlighting the prosthetic components predisposed to wear. We have added this additional relevant literature to supporth the explanation (lines 38-42).

  1. In the materials and methods, the authors need to add additional illustrations as a form of figure that explains the workflow of the present study to make the reader easier to understand rather than only the dominant text as a present form.

The flow chart of the patient selection and inclusion process has been added in the revised manuscript. The figure shown all patients identified, the potential eligible patients for each revision surgery and those excluded. The illustration contains also the total number of patients included and it has been introduced in the result  Figure1 and (lines  1456-173)

  1. What is the baseline of patient selection? Is there any protocol, standard, or basis that has been followed? It is unclear since the patient is very heterogeneous with a small number. The resonance involved impacts the present result makes this study flaws. One major reason for rejecting this paper.

The present retrospective observational case-control study used medical file records of patients included in a registry of orthopaedic surgical procedures. All patients signed an informed consent to be included in the registry. All surgeries were performed by senior surgeons trained in joint replacement surgery at the Humanitas Research Hospital IRCCS, Italy. Patients included were divided into two groups based on age at the time of surgery: < 80 years old (Group 1) and those geriatrics ≥ 80 years old (Group 2). Participants exclusion criteria were sequelae of periprosthetic infection, malignancy, patient under 18 years old and less than 1 year of follow up (lines 69-79).

  1. It's also essential to include additional information on the manufacturer, country, and specifications of the tools.

All surgeries were performed by senior surgeons trained in joint replacement surgery at the Humanitas Research Hospital IRCCS, Italy. Femoral stem revision was performed using undemented monoblock splined tapered gritblasted stems (Wagner-SL stem; Zimmer, Warsaw, IN) or uncemented Arcos Modular Femoral Revision System (Zimmer Biomet Inc., Warsaw, IN) in case of major defect in order to avoid leg length discrepancy LLD). Acetabular revision was performed using Trabecular Metal shell (TM Modular, Trilogy TM and augment or TM Revision shell, Zimmer Biomet, Warsaw, IN), for major acetabular defect dual mobility cup (DMC Avantage; Zimmer Biomet, Warsaw, IN) cemented into a larger shell was used. Revision knee arthtroplasty was performed using NexGen® Legacy® Constrained Condylar Knee (LCCK) and Trabecular Metal™ Cones augments (Zimmer Biomet, , Warsaw, IN) to help provide stabilization.

10.The error and tolerance of the experimental tools used in this investigation are important aspects that have to be mentioned in the manuscript. It might be valuable for further research by other scholars because of the different results.

The present study is a retrospective observational case-control study that used medical file records, as with any database the quality of data and missing data may introduce errors. However, we included patients whom we had all data for primary and secondary endopoints.

  1. Results must be compared to similar past research.

We introduced new studies to enhance discussion and compare results.

  1. The authors need to improve the discussion in the present article become more comprehensive. The present form was insufficient.

We improved the discussion comparing results to past research and introducing limitation of the study (lines 319-330).

  1. Please include the limitation of the present study, it is missing.

We reported the limitations and strength of the study in the discussion (lines 319-330).

  1. Information to the conclusion by formatting it as a paragraph rather than the manner as point-by-point.

The conclusion has been as a paragraph (lines 331-337).

  1. The conclusion section needs to explain further research.

The conclusions end with the need for future research (lines 335-337).

  1. The authors need to enrich the reference from five years back. MDPI reference is strongly recommended.

We have enriched the reference with recent publications. These have been helpful to compare the results to similar past researchs.

  1. The authors were encouraged to proofread their work due to grammatical problems and linguistic style.

We have accurate proofread the manuscript, we hope that quality has improved.  

  1. It is suggested to the authors for providing graphical abstract in the system after revision.

Thanks for this suggestion, we added a graphic abstract alongside the text abstract as required.

We thank the Editorial Board and Reviewers for revising our manuscript. We appreciate your and the reviewer’s comments. We hope that the overall quality of the manuscript has improved, and is now amenable for publication in the Journal of Clinical Medicine.

Reviewer 2 Report

Thank you for giving me the possibility to review the manuscript "Perioperative complications after hip and knee revision arthroplasty in the over 80 years old population: a retrospective observational case-control study”.
The paper deals with a very interesting topic in orthopaedic surgery since it aims to evaluate the safety of hip and knee revision arthroplasty in patients over 80 years old. The primary outcome was to determine and compare the incidence of in-hospital perioperative medical and surgical complications, both local and systemic, following THA and TKA revisions of patients over 80 years old compared to the younger control group. The secondary outcome was to determine and compare the 30-day, 90-day, and 1-year readmission rates between these two groups.
The paper is well-written, the methods are complete, the statistical analysis is well-conducted, and the potentials and limitations of the study are clearly reported.
However, before considering it for publication in JCM, the following points should be addressed:
1 please improve the quality of the figures.

Author Response

Manuscript ID: jcm- 2160996

Type of manuscript:

Article Title: Perioperative complications after hip and knee revision arthroplasty in the over 80 years old population: a retrospective observational case-control study.

Dear Reviewer 2,

Please find attached the revised version of the above manuscript. Your comment of the reviewers have been carefully considered, and implemented as follows. Please note that all changes made in the document have been highlighted in yellow to facilitate tracking and reading.

Reviewer 2

Thank you for giving me the possibility to review the manuscript "Perioperative complications after hip and knee revision arthroplasty in the over 80 years old population: a retrospective observational case-control study”. The paper deals with a very interesting topic in orthopaedic surgery since it aims to evaluate the safety of hip and knee revision arthroplasty in patients over 80 years old. The primary outcome was to determine and compare the incidence of in-hospital perioperative medical and surgical complications, both local and systemic, following THA and TKA revisions of patients over 80 years old compared to the younger control group. The secondary outcome was to determine and compare the 30-day, 90-day, and 1-year readmission rates between these two groups.
The paper is well-written, the methods are complete, the statistical analysis is well-conducted, and the potentials and limitations of the study are clearly reported.
However, before considering it for publication in JCM, the following points should be addressed:
1. please improve the quality of the figures.

We have removed the poor quality graphics. We have drawn up new graphics and added them in the article in order to improve the definition of the figures (Figure 2, Figure 3 and Figure 4).

We thank the Editorial Board and Reviewers for revising our manuscript. We appreciate your and the reviewer’s comments. We hope that the overall quality of the manuscript has improved, and is now amenable for publication in the Journal of Clinical Medicine.

Round 2

Reviewer 1 Report

Many acknowledge to the authors for their revision, several followed comments as response for the revised version needs to be addressed as follows:

1.      the abbreviations THA and TKA are given in line 35, but previously no mention of their abbreviations. Please present it in a revised version.

2.      Medical aspect having directly related with mechanical aspect of the implant, that making the mechanical aspect become crucial for avoiding medical problem to user. I am encouraged the authors for discuss it for extending explanation. Relevant reference for support the explanation as follow: The Effect of Bottom Profile Dimples on the Femoral Head on Wear in Metal-on-Metal Total Hip Arthroplasty. J. Funct. Biomater. 2021, 12, 38. https://doi.org/10.3390/jfb12020038

3.      As stated in line 112-122, please give explanation regarding the basis of performed statistical analysis.

Author Response

Manuscript ID: jcm- 2160996

Type of manuscript:

Article Title: Perioperative complications after hip and knee revision arthroplasty in the over 80 years old population: a retrospective observational case-control study.

Dear Reviewer 1,

Please find attached the revised version of the above manuscript. Your comments have been carefully considered, and implemented as follows. Please note that all changes made in the document have been highlighted in yellow to facilitate tracking and reading.

Reviewer 1

  1. the abbreviations THA and TKA are given in line 35, but previously no mention of their abbreviations. Please present it in a revised version.

We reported THA (total hip arthroplasty) and TKA (total knee arthroplasty) abbreviations (lines 36-37).

  1. Medical aspect having directly related with mechanical aspect of the implant, that making the mechanical aspect become crucial for avoiding medical problem to user. I am encouraged the authors for discuss it for extending explanation. Relevant reference for support the explanation as follow: The Effect of Bottom Profile Dimples on the Femoral Head on Wear in Metal-on-Metal Total Hip Arthroplasty. J. Funct. Biomater. 2021, 12, 38. https://doi.org/10.3390/jfb12020038

 We discussed the relation between mechanical aspect of the implant and medical aspect.  We have enriched this additional relevant literature to supporth the explanation and others necessary for discuss the extending explanation (lines 270-282)

  1. As stated in line 112-122, please give explanation regarding the basis of performed statistical analysis.

The Association between risk factors and early mobilization at day 0 and risk factors and postoperative anaemia were explored with univariable logistic regression. All independent factors with a p value under 0.1 were then submitted to a backward multivariable logistic regression. An explorative analysis was performed for association of age groups and for systemic complications, 30-day, 90-day and 1-year readmission rate. A p value under 0.05 was considered as significant. All analyses were performed with stata15.

We thank the Editorial Board and Reviewers for revising our manuscript. We appreciate your and the reviewer’s comments. We hope that the overall quality of the manuscript has improved, and is now amenable for publication in the Journal of Clinical Medicine.